# Variability of Hydraulic Properties and Hydrophobicity in a Coarse-Textured Inceptisol Cultivated with Maize in Central Chile

**Nicolás Riveras-Muñoz** [1,2,*] , **Carla Silva** [1], **Osvaldo Salazar** [1] , **Thomas Scholten** [2] , **Steffen Seitz** [2] and **Oscar Seguel** [1]

1    Departamento de Ingeniería y Suelos, Facultad de Ciencias Agronómicas, Universidad de Chile, Santiago 8820808, Chile
2    Department of Geosciences, Soil Science and Geomorphology, University of Tübingen, Rümelinstrasse 19–23, 72070 Tübingen, Germany
*    Correspondence: nicolas-andres.riveras-munoz@uni-tuebingen.de

**Abstract:** The O'Higgins Region, located in Central Chile, concentrates 40% of the country's maize production, mainly under conventional tillage. This has generated soil physical degradation, modifying water movement on it, which varies even in short distances. In this study, we wanted to evaluate the spatial variability of different physical and hydraulic properties in relation to the agricultural use of Inceptisol. The study was conducted on a farm in Central Chile, in a fallow–maize rotation under conventional tillage. Penetration resistance (PR) was measured by using systematic sampling, defining areas of high and low PR, where soil samples were collected in places with frequent crossing of machinery (+M) and places without crossing (−M) and on topsoil and subsoil, establishing four treatments: +M—topsoil, +M—subsoil, −M—topsoil, and −M—subsoil. Organic matter (OM), texture, bulk density (BD), hydraulic conductivity (K), and hydrophobicity (R) were measured. The soil order was Inceptisol with a sandy-loam texture. The PR ranged from 200,000 to 2,000,000 Pa on topsoil and 600,000 to 2,400,000 Pa in subsoil, and the OM content was higher with a low PR. The K varied from 0.6 to 18 cm h$^{-1}$, being greater in depth, as tillage disturbs the topsoil stabilized during the season. A linear relationship was found between the K and R, explaining differences between high- and low-PR sites. There was an association between $K_{sat}$ with position (subsoil/topsoil) and PR (high/low) that may allow us to use the PR as a proxy for K.

**Keywords:** soil physical functioning; hydraulic conductivity; conventional tillage; penetration resistance; repellency index

## 1. Introduction

One of the main economic activities in Central Chile is agriculture, with maize production at the first position with 78% of the total crop surface and indicating O'Higgins Region as the largest agricultural planting area at national level [1]. The average regional yield of 11.7 Mg ha$^{-1}$ is above the national average of 10.6 Mg ha$^{-1}$ but below the potential yield of 22.8 Mg ha$^{-1}$ that Pioneer [2] has reported.

While Central Chile shares with other Mediterranean regions dominance of cloudless skies during the growing season, it has two advantages for maize production: (i) the cooling influence of the Humboldt Current that flows north along the western coast of South America, providing cool temperatures in summer and promoting the accumulation of biomass; and (ii) in most of the world, maize is grown as a rainfed crop, while in Chile and some other very limited regions, it is irrigated, resulting in yields far above the world average and making it difficult to make a straightforward comparison with other regions [3].

Despite the very favorable conditions, the gap with the potential yield indicates the presence of limitations. In this sense, Taylor and Brar [4] point out root decreases and soil compaction as one of the main issues affecting the yield of maize, while Hamza and Anderson [5] attribute hydraulic properties, such as reduction in infiltration and available water, as determining factors in crop yield and root growth. In this regard, maize farmers associated with the Cooperativa Campesina Intercomunal de Peumo (COOPEUMO), in the context of a Clean Production Agreement (CPA), indicated the presence of compaction and surface runoff problems, likely related to long-term monoculture and intensive tillage [6].

Tillage represents one of the most influential disturbances of the soil surface, resulting in changes of varying intensity for different soil properties [7]. Although the general physical restrictions generated by overcultivation are well studied, their characterization and applied assessment is difficult due to their unknown spatial distribution and high variability [8]. Soil mechanical and hydraulic processes are interrelated and affect each other. Therefore, when the soil structure and pore distribution are modified, water retention and hydraulic conductivity (K) should change [9]. The magnitude of these changes depends on external factors, such as (i) the duration and frequency of the tillage operations and (ii) the compaction produced by the transit of agricultural machinery and/or animal trampling [10].

Sandy soils have a high K at low pressures (close to saturation), due to their greater amount of macropores (>10 μm), provided by the space between the coarse particles. However, when the pores are quickly depleted, they are filled with air, thus resulting in a sharp decrease in the water conduction, as reflected in low values of unsaturated hydraulic conductivity ($K_h$) [11].

A widely spread and sustainable way to avoid those harmful effects on the soil in the agricultural production is the addition of organic matter. Organic matter stabilizes the contact points between soil particles through cementing and the hydrophobic effect, which prevents solutes from being dissolved by water [12], and it is the main structuring agent in the case of coarse texture dominated soils [13].

However, due to the physicochemical characteristics of certain organic compounds that coat the solid particles and have a hydrophobic effect, it can result in a reduction of the wetting rate and water repellency in the soil [14]. Cuevas Becerra [15] describes this phenomenon as a process of high relevance since its existence is associated with preferential flows and surface runoff.

Finally, it has to be considered that the long-term repeated tillage and incorporation of organic residues into cereal production systems leads to compaction problems that can hinder the capacity of fluid transmission in the soil [16]. Additionally with changes in local hydrology, related to the effect of hydrophobicity and the destruction of the porous system by farming, where its distribution and variability are unknown [17], we hypothesized that, in a coarse textured soil under long-term tillage and maize monoculture, the penetration resistance and its spatial distribution are directly related to the hydraulic conductivity and hydrophobicity of the soil. The aims of the study were (i) to quantify the mechanical resistance of the soil and its distribution both vertically and horizontally in areas of high and low mechanical resistance, places with frequent crossing and places without crossing of machinery, such as at the topsoil and subsoil; and (ii) evaluate the hydraulic conductivity and hydrophobicity depending on the spatial variability of the soil and its dependence on soil compaction for a coarse textured soil under a long-term tillage and maize monoculture.

## 2. Materials and Methods

### 2.1. Site Description

The study was carried out during the 2013/2014 season, on a 2.9 ha site belonging to an associate of the Cooperativa Campesina Intercomunal de Peumo (COOPEUMO), located at San Luis in the Central Valley of Chile, O'Higgins Region, Commune of Pichidegua (Figure 1). The soil was classified as Typic Haploxerepts (Inceptisols after US Soil Taxonomy) on alluvial terraces [18], with textural classes varying from mainly loam at the surface to

sand at 30 cm depth. Topsoil (0–15 cm) is characterized by its neutral soil pH (i.e., 6.93), low OM content (OM = 1.47%), non-saline quality (EC = 1.59 dS m$^{-1}$), and low cation exchange capacity (CEC = 9.65 cmol$_{(+)}$ kg$^{-1}$), according to Salazar, et al. [19].

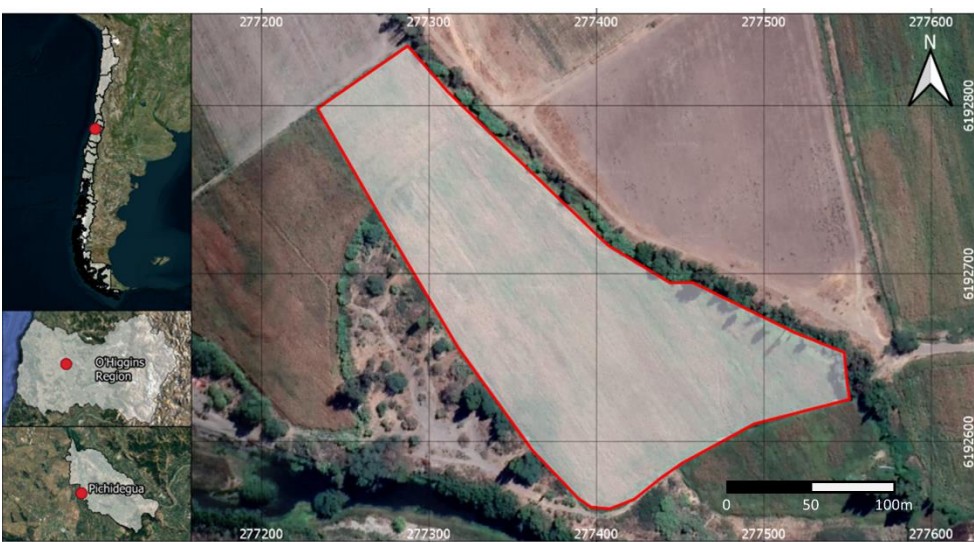

**Figure 1.** Location map of San Luis (277399 E° 6192699 S), experimental site (Google Earth, datum WSG 1984).

The Commune of Pichidegua is under a semi-arid Mediterranean climate. It has an average annual air temperature of 14.6 °C, a monthly maximum of 26.3 °C in the month of January, and a monthly minimum of 6.1 °C in August. Precipitations are concentrated from May to October and average about 550 mm year$^{-1}$, with a potential evapotranspiration of 940 mm year$^{-1}$ [20].

At the study site, long-term tillage and maize monoculture has been carried out (>15 year). The soil is prepared by using a disc plough in September, and maize is sown in October and harvested in early April/May. During the growing season, 470 kg N ha$^{-1}$ was applied as urea and compound fertilizer (N–P$_2$O$_5$–K$_2$O: 25–10–10), and a N balance estimated at 200 kg N ha$^{-1}$ surplus was available for nitrate leaching. Maize production is performed under furrow irrigation with conventional tillage management, as described by Salazar, et al. [21]. Soil preparation with a disc plough is carried out immediately after the harvest or before planting, with the number of tractor passes varying from 5 to 8 times, using the same routes, making it possible to identify places with frequent crossing of machinery (+M) and places without the crossing of it (−M). The grain yield in 2012 was 15 Mg ha$^{-1}$. Stubble and plant residues from the previous season are consumed by cattle through direct grazing, and the remnant is incorporated into the soil.

### 2.2. Study Design

In November 2013, with the maize crop in eighth leaf, an initial diagnosis of the state of compaction of the study site was performed. Soil penetration resistance (PR) was measured by identifying areas of high and low PR (according to the 0–5 cm layer). In each zone, four experimental units of 1 m$^2$ each were delimited randomly (n = 8). Within each experimental unit, the zone related to the passage of machinery was identified, with places of frequent crossing of it (+M) and places with no crossing of machinery (−M). Moreover, due to repeated handling over several seasons, based on visual characteristics and mechanical resistance, it was possible to identify the limit of the plow layer (approximately 30 cm), using these criteria to separate in topsoil and subsoil layers. From the combination of the tire tracks of the machinery and the limit of the arable layer, four treatments were established: T1: −M—topsoil; T2: −M—subsoil; T3: +M—topsoil; T4: +M—subsoil. In each PR zone, the K was measured on the field for all four treatments, with four replicates (n = 32). Complementary to each

K measurement, we collected undisturbed soil cylinders (heigh = 5 cm, diameter = 5.9 cm, volume = 136.7 cm$^3$) for the determination of hydrophobicity, disturbed soil samples at depths of 0–10 cm and 30–40 cm, and three disturbed cylinders for BD.

*2.3. Soil Property Measurements*

Soil compaction was characterized as the spatial variability of penetration resistance (PR) with a Penetrologger (Eijkelkamp, Giesbeek, The Netherlands), with measurements distributed in a regular 30 m × 30 m grid. All the measurements were taken in the same position of the furrow, avoiding the tractor tread, to prevent distortions of the values. The device was equipped with a GPS and data storage memory, allowing us to spatially locate the evaluated points.

Considering that mechanical loads exert their greatest effect in the topsoil layer, PR mean values from 0 to 5 cm were interpolated by kriging, generating a spatial distribution map of PR. The measurement was performed 2 days after an irrigation, with a water content close to field capacity and a penetration down to 80 cm in depth. The results were used to identify two classes of penetration resistance (high and low), using the median PR as threshold.

Texture was measured according to the Bouyoucos hydrometer method, bulk density (BD) was determined by the cylinder method [22], and the content of organic matter (OM) was determined by calcination [23].

Unsaturated hydraulic conductivity ($K_h$) measurements were conducted in the field between April and May 2014 and prior to farmers' soil preparation for the next season. It was measured with mini disk infiltrometers (Decagon Devices, Pullman, WA, USA). A fine sand layer was added to ensure the contact of the porous plate to the ground. The infiltrated water was measured every 30 s for a period of 10 min, for 100, 200, 400, and 600 Pa of soil water pressure. The obtained data were processed by the method of Zhang [24] to determine $K_h$ as function of the applied soil water pressure, defined as follows:

$$K_h = C_1 / A \tag{1}$$

where $C_1$ corresponds to curvature of the parabola at pressures of 100, 200, 400, and 600 Pa, obtained from a quadratic regression between the accumulated infiltration and the root of time, and A is a factor relating the van Genuchten parameters [25] and depends of soil texture, the sorption rate, and the radius of the infiltrometer disk [26].

With the distribution of $K_h$ as a function of the supplied pressure, linear adjustments were made to extrapolate the saturated hydraulic conductivity ($K_{sat}$) to 0 Pa pressure.

Hydrophobicity was evaluated by the repellency index (R) according to the methodology described by Tillman, et al. [27], using a self-made sorptivity device based on the specifications of Leeds-Harrison, et al. [28]. The device consists of a network of 4 mm capillaries that conduct the liquid from a container to an air-dry soil sample in close contact to a sponge, setting at the end a negative pressure (h) of 1 cm of water column or −100 Pa, which causes a suction that makes the liquid flow through the capillary. The amount of liquid held in the sample and the capillary system was registered with a precision balance, and the R index was obtained from the ratio of water and ethanol 95%.

First the test was performed by infiltrating water; then the sample was air-dried, and finally the ethanol infiltration test was repeated. The infiltration of each liquid was measured every 15 s, until 75 s, and from the density of each liquid, the final infiltration volume was determined. With the data recorded, the volume of water was plotted as a function of time, obtaining the liquid flow rate. From this, we estimated the sorptivity in water and ethanol, considering a capillary of 4 mm diameter, factor b = 0.55, and f = 1.0, according to the proposed by Hallett and Young [29]. We determined the index R by using the following equation:

$$R = 1.95 \times (S_e / S_w) \tag{2}$$

where $S_e$ corresponds to sorptivity in ethanol, $S_w$ to sorptivity in water, and 1.95 is due to the constant that considers the properties of water and ethanol (viscosity and surface tension).

Soils were considered hydrophobic if R was greater than 1.95 and hydrophilic if it was less than 1.95 [27].

*2.4. Statistical Analysis*

First, a penetration resistance (PR) map was generated to determine the distribution of this property in the study site. For this, a linear interpolation with ordinary kriging was made in R, using the package gstat 2.0–9 [30]. Areas of high PR and low PR were calculated based on the median (485,000 Pa) of the PR as threshold value and then used as blocking criteria.

A randomized complete block design (RCBD) was established, using areas with high and low PR as blocking criteria. In each block, a 2-factorial treatment was applied, from the combination of (i) two relative positions to the tractor's track (places with frequent crossing of machinery (+M) and places without crossing (−M)) and (ii) two relative positions to the plow layer (topsoil and subsoil), with a total of 4 treatments. Four repetitions of each treatment were randomly distributed in each block.

To determine the treatment effect, an ANOVA was conducted with a confidence level of 95%, using the Infostat software [31]. In the case of significant differences, the treatments were analyzed with the test of multiple comparisons of the Least Significant Difference (LSD) ($\alpha \leq 0.05$). The results of hydraulic conductivity (K) were determined based on a linear regression of $K_h$ by the supplied tension (100, 200, 400, and 600 Pa). Then the regression coefficients were compared through a *t*-test, comparing pairs with the slopes and intercepts of the linear model. Moreover, a generalized linear mixed model (GLMM) was performed with the whole set of data ($K_h$ for four repetitions at four soil water pressure stages, n = 16) and normalized distributions when the data did not show normal distribution, and therefore the assumptions for errors were not fulfilled [32]. Because both approaches gave comparable results, the first statistical analysis is presented as representative of the $K_h$, but GLMM was used to determine the confidence intervals for intercepts (95%), equivalent to $K_{sat}$.

Dependence between variables was explored through stepwise regression, using the stats package of R 3.5.3 [33]. Model selection was made based on simplicity, Akaike Information Criterion (AIC), and avoiding collinearity between variables.

Finally, a correlation analysis was performed between hydrophobicity (R index) and saturated hydraulic conductivity ($K_{sat}$) to determine if the water repellency alters the water flow in the soil. For this, a logarithmic adjustment of the variable $K_{sat}$ was performed as a function of the R index, achieving an adjustment with a significance of 95%.

## 3. Results

*3.1. State of Soil Compaction and Related Physical Soil Properties*

3.1.1. Penetration Resistance (PR)

When looking at how the PR behaves vertically (Figure 2), it increases up to approximately 40 cm, which coincides with the plowing depth. A clear difference of approximately 400,000 Pa can be seen between the low and high-PR zones, with higher values in the latter. More profound than 40 cm, PR tends to decrease in depth, with heterogeneous but similar values between high and low PR.

The areas of high and low PR can be clearly identified on the spatial distribution map of PR obtained in field (Figure 3). Low-PR zones range from 156,000 to 485,000 Pa, which corresponds to the median of the values between 0 and 5 cm, while high-PR zones range from 485,000 to 263,000 Pa. The high-PR zone coincides with the entrance into the field, which presents higher values of penetration resistance, where continuous traffic has caused a deterioration of the soil structure and the porous system [34].

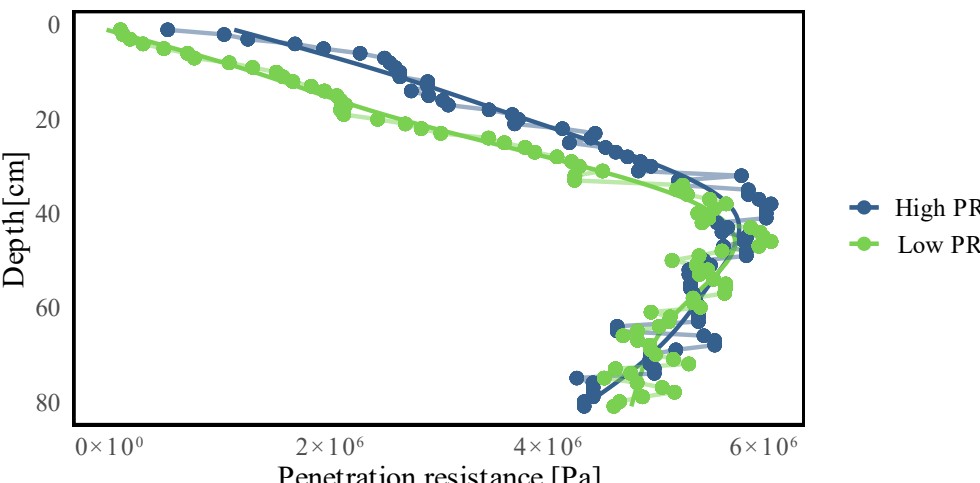

**Figure 2.** Average values of high (blue) and low (green) PR along the soil profile, defined in the 0–5 cm soil layer.

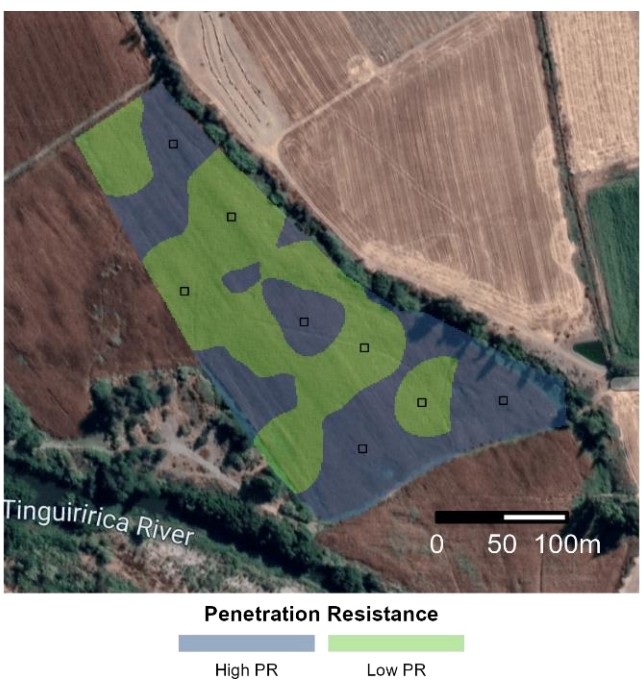

**Figure 3.** Spatial distribution of PR (0–5 cm) at the study site. Areas of high PR (blue) and low PR (green). Sampling sites are indicated with a black square. Cutoff threshold corresponds to the median value (485,000 Pa).

### 3.1.2. Texture

The granulometric analysis of the soil allowed us to define a class of sandy-loam texture (SL) for the whole study site, with an average percentage of clay of 7.4%. It was a trend of increasing sand content and decreasing silt content with depth. In addition, there were differences for clay and sand between the high- and low-PR areas (Table 1). The high-PR area presented a higher average value of sand and lower average value of clay than the zone of low PR. Even so, within each analyzed area, there is a high textural homogeneity which allows us to compare the treatments as a result of management.

**Table 1.** Granulometry at the topsoil (0–10 cm) and subsoil (30–40 cm) for each area of the study site, places with frequent crossing of machinery (+M), and places without crossing (−M). Mean ± SD.

| Zone | Treatment | Position | Depth | Texture [a] | | | Textural Class |
| | | | | Sand | Silt | Clay | (USDA) [b] |
| | | | | ————(%)———— | | | |
| High PR | T1 | −M | topsoil | 57.6 ± 2.2 | 34.9 ± 1.9 | 7.5 ± 1.8 | SL |
| | T2 | −M | subsoil | 64.7 ± 2.1 | 28.8 ± 17.2 | 6.5 ± 3.9 | SL |
| | T3 | +M | topsoil | 57.0 ± 5.6 | 35.9 ± 10.2 | 7.2 ± 2.2 | SL |
| | T4 | +M | subsoil | 64.6 ± 15.9 | 30.5 ± 9.2 | 4.9 ± 0.6 | SL |
| | | | | 61.0 ± 4.3 a | 32.5 ± 3.4 a | 6.5 ± 1.1 b | SL |
| Low PR | T1 | −M | topsoil | 51.8 ± 5.1 | 40.4 ± 2.9 | 7.9 ± 2.7 | SL |
| | T2 | −M | subsoil | 57.0 ± 7.9 | 35.2 ± 6.1 | 7.8 ± 2.1 | SL |
| | T3 | +M | topsoil | 50.5 ± 2.1 | 40.4 ± 2.8 | 9.2 ± 1.3 | SL |
| | T4 | +M | subsoil | 55.2 ± 4.9 | 36.4 ± 4.3 | 8.4 ± 0.9 | SL |
| | | | | 53.6 ± 3.0 b | 38.1 ± 2.7 a | 8.3 ± 0.6 a | SL |

[a] Different letter between averages of high- and low-PR areas for the same particle size denote statistically significant differences ($p < 0.05$). [b] SL: sandy loam.

### 3.1.3. Bulk Density (BD)

BD values in general show great variability, so no significant differences attributable to treatments were obtained (Figure 4). Although a higher BD was expected in places with frequent crossings than in non-crossed areas, this was only true for the high-PR areas. Slightly higher values were observed in the subsoil. The areas of low PR showed lower values of BD in places with frequent crossing (+M) compared to places without crossing (−M). However, these values were measured approximately 6 months after tillage, so they were probably higher than those recorded immediately after soil preparation.

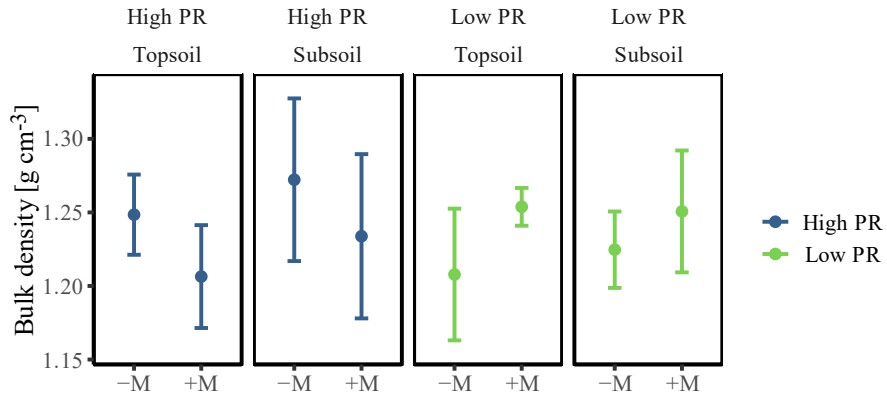

**Figure 4.** Bulk density (BD) as function of the treatments of the study. Average ± standard deviation values. There were no statistically significant differences ($p > 0.05$).

### 3.1.4. Soil Organic Matter (OM)

In this study, the OM contents differed between the treatments in different PR zones (Figure 5). When comparing high- and low-PR areas, the high-PR sector showed a lower level of OM of 1.78 ± 0.48% than the low-PR area, with 2.23 ± 0.30% ($\alpha \leq 0.05$). On average, the low-PR area had 0.45% more OM than the high-PR site, equivalent to approximately 22 Mg ha$^{-1}$ organic C between 0 and 40 cm depth. At the study site, practices of incorporation of crop residues are frequently carried out, and mechanical chipping has replaced the burning of stubble. However, in the present study, the recorded contents of OM did not show great variability among treatments and presented levels which are considered to be low (<2.5%).

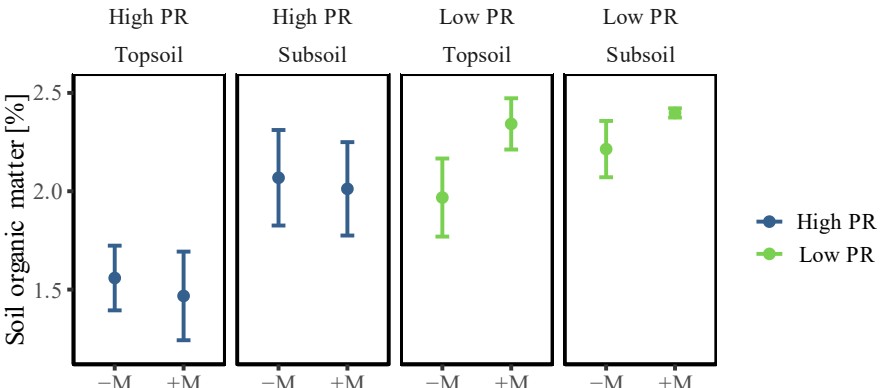

**Figure 5.** Organic matter content (OM in %) as a function of position and depth. Average ± standard deviation values. In each zone, there were no statistically significant differences ($p > 0.05$) between treatments.

*3.2. Hydraulic Functionality of the Soil*

3.2.1. Unsaturated Hydraulic Conductivity ($K_h$)

Casanova, et al. [35] proposed to describe the behavior of the $K_h$ as a function of the water pressure with an adjustment of two straight lines: a first one describing the behavior of $K_h$ close to the minimum size of mesopores and a second line describing the behavior close to saturation, associated with macropores, with the latter being more pronounced in clay soils. In the present study, given the textural class of the soil and the low water supply pressures, the behavior within the first section of rectilinear character was considered (Figure 6), allowing us to extrapolate the saturated hydraulic conductivity ($K_{sat}$) by using the value of zero pressure by linear adjustment (Table 2).

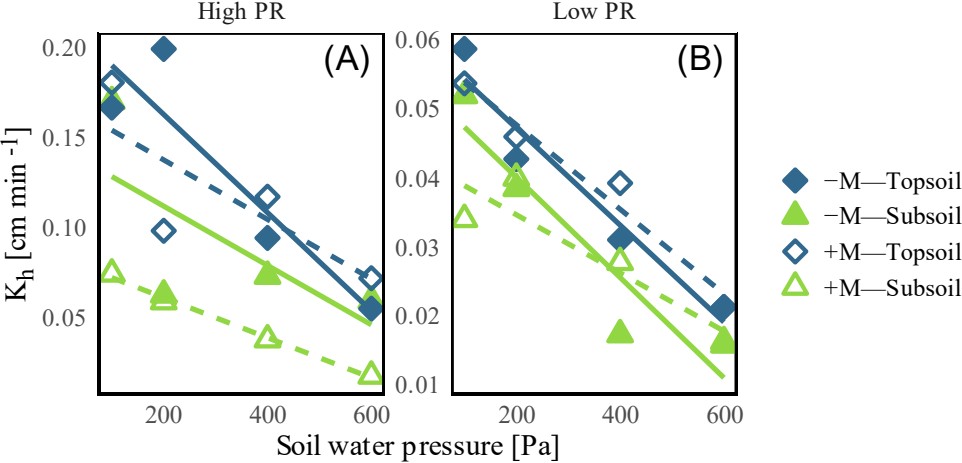

**Figure 6.** Unsaturated hydraulic conductivity ($K_h$) measured in the soil at pressures of 100, 200, 400, and 600 Pa. (**A**) High-PR area and (**B**) low-PR zone. The legend applies to both figures. Note the scale change of the *y*-axis.

The average $K_h$ obtained in the field showed that it was greater in the subsoil than in the topsoil, as there is a higher sand content at a greater depth (Table 1). In addition, at the time of taking the measurements, the formation of a laminar layer of fine particles that sealed the soil against the entry of water was observed on the surface.

The linear regression coefficients (Table 2) of the relation between soil water pressure and unsaturated hydraulic conductivity (Figure 6) present a negative slope, indicating an inverse relation between those variables. However, due to the high variability, not all cases fit the rectilinear model well. The slope and the intercept are greater in the places with frequent crossing of machinery (+M) than in the places without crossing of it (−M) at the

same depth, more notoriously in the high-PR zone. The saturated hydraulic conductivity can be obtained from the extrapolation of soil water pressure to 0 Pa. It maintains higher values in the subsoil for the +M condition, as statistically validated by *t*-tests.

**Table 2.** Linear model and its coefficients for unsaturated hydraulic conductivity ($K_h$) as a function of the supply pressure (Pa) for the averages of the replicates. Different letters indicate statistically significant differences (*p* <0.05) according to the *t*-test.

| Zone | Treat. | Position | Depth | $K_h$ (cm min$^{-1}$) | *p*-Value | *t*-Test |
|------|--------|----------|-------|----------------------|-----------|----------|
| High PR | T1 | −M | topsoil | T1 = −0.017x + 0.145 | 0.1459 | ab |
| | T2 | −M | subsoil | T2 = −0.027x + 0.218 | 0.0065 | a |
| | T3 | +M | topsoil | T3 = −0.011x + 0.083 | 0.0003 | b |
| | T4 | +M | subsoil | T4 = −0.017x + 0.171 | 0.1693 | ab |
| Low PR | T1 | −M | topsoil | T1 = −0.007x + 0.055 | 0.0409 | ab |
| | T2 | −M | subsoil | T2 = −0.007x + 0.062 | 0.0007 | a |
| | T3 | +M | topsoil | T3 = −0.004x + 0.043 | 0.1329 | b |
| | T4 | +M | subsoil | T4 = −0.006x + 0.060 | 0.0323 | ab |

### 3.2.2. Saturated Hydraulic Conductivity ($K_{sat}$)

As expected, the $K_{sat}$ presented values that increased in depth, given the increase in both the content and the size of the sands (Table 3). The $K_{sat}$ collected data show no effect on the places with frequent crossing of machinery (+M) and places without crossing (−M), but a significant decrease of it in low PR and topsoil, related to their respective complements. The $K_{sat}$ values are considered to be high, since they are within the range of 0.06 to 0.6 cm min$^{-1}$.

**Table 3.** Saturated hydraulic conductivity (mean, upper limit, and lower limit derived by GLMM) as a function of depth in high- and low-PR areas and for sites frequently crossed (+M) or non-crossed (−M) by machinery. Confidence intervals have been computed with a 95% confidence level.

| Position | Depth (cm) | $K_{sat}$ (cm min$^{-1}$) High PR (UL; LL) | Low PR (UL; LL) |
|----------|-----------|------------------------------------------|------------------|
| +M | topsoil | 0.083 (−0.021; 0.191) | 0.043 (−0.028; 0.116) |
| | subsoil | 0.171 (0.069; 0.281) | 0.060 (0.031; 0.092) |
| −M | topsoil | 0.144 (0.073; 0.224) | 0.055 (0.005; 0.107) |
| | subsoil | 0.218 (0.117; 0.329) | 0.061 (−0.009; 0.135) |

The confidence intervals showed a great dispersion (Table 4). Consequently, the $K_{sat}$ values behave independently of the positions and depths at which they were measured (Table 3).

**Table 4.** Repellency index (R, dimensionless) obtained by the method proposed by Tillman, et al. [27] in air-dried samples.

| Zone | Treat. | Position | Depth (cm) | R-Index |
|------|--------|----------|-----------|---------|
| High PR | T1 | −M | topsoil | 1.33 (±0.37) |
| | T2 | −M | subsoil | 1.57 (±0.38) |
| | T3 | +M | topsoil | 1.64 (±0.42) |
| | T4 | +M | subsoil | 1.39 (±0.40) |
| Low PR | T1 | −M | topsoil | 1.70 (±0.63) |
| | T2 | −M | subsoil | 1.58 (±0.69) |
| | T3 | +M | topsoil | 2.02 (±0.70) |
| | T4 | +M | subsoil | 1.70 (±0.26) |

### 3.2.3. Water Repellency Index (R)

Based on a previous diagnosis [6] and because water repellency tends to increase in coarse-textured soils [36], we expected to find this phenomenon at the study site. Nevertheless, the treatments did not show water repellency or hydrophobicity (Table 4), with the exception of Treatment 3 of the low-PR area, with an R value slightly higher than 2. However, there were no differences ($p > 0.05$) between treatments or between high- and low-PR areas.

## 4. Discussion

### *4.1. State of Soil Compaction and Related Physical Soil Properties*

#### 4.1.1. Penetration Resistance (PR)

The knowledge of the spatial distribution of soil penetration resistance is useful to identify compacted areas. With this problem stated, management measures can be generated to minimize negative effects, such as decrease in crop yields or the risk of erosion due to increased surface runoff [37]. According to our results, the identification of the variability of the soil mechanical resistance allowed us to determine two zones: high PR and low PR (based on 0–5 cm depth). Accordingly, we then performed evaluations of the other soil properties for each zone separately to obtain homogeneous treatments and to decrease the variability between the samples.

The area of high PR near the entrance to the field is a consequence of repeated and concentrated traffic at the site and the long-term use of it in maize production. This continuous use has also caused a deterioration of the soil structure and the porous system [34] with a reorganization of the particles, generating a plow pan of approximately 40 cm in depth. The plow pan also leads to a change of the general trend, from a clear differentiation of the PR in the topsoil to a homogenization in the subsoil, that can be explained by the stratification product of the compaction by mechanization [38]. Even if the PR does not exceed 6,000,000 Pa, it is considered to be at a very high level according to Schoeneberger, et al. [39]

The variability of the surface PR zone can be explained by the natural variability within the textural classes of the soil that is intervening in the response to compaction processes (cf. Horn [40]; see below).

It is known that the effect of the number of passages of a tractor is cumulative and that the PR increases with the intensity of the traffic [10]. This was evaluated by Usowicz and Lipiec [37], who determined that the maximum effect occurs between the first and third passage of the tractor at a depth range of 0–60 cm and that this effect decreases to a greater depth. Thus, consequences are manifested predominantly near the soil surface. In addition, Horn and Fleige [41] showed that tires with a load of 3.3 Mg exert an average normal stress of approximately 50,000 Pa to the 40 cm depth layer, and increasing this load to 6.5 Mg leads to tensions of more than 100,000 Pa, even down to 60 cm in depth. The same authors clarified that a loamy soil presents a greater sensitivity to the external loads than a sandy or clayish soil due to differences in the pore size distribution.

The effects on root growth as result of increased mechanical strength were studied by Taylor and Brar [4], who determined a linear decrease in root elongation with PR values higher than 500,000 Pa. On the other hand, Pérez, et al. [42] considered that root growth equals zero with a soil penetration resistance of approximately 5,000,000 Pa, and, in the specific case of the maize crop, Hadas [43] reported no root growth values in a range of 1,600,000 to 3,700,000 Pa. Feldman [44] indicated that the typical rooting depth of maize varies between 150 and 180 cm, depending on the textural class of the soil. This finding indicates that, at the sites of low PR, the rooting of maize will be affected under 10 cm of depth (Figure 2), resulting in low exploration of the plant roots and reduction and heterogeneity of the maize yields at the study site.

#### 4.1.2. Texture

The texture influences the mechanical behavior and ease of soil tillage, the amount of water and air it can hold, and the speed with which water penetrates and moves in it [45].

The particle size is determined by the parent material and the pedogenesis, accordingly affecting erosion processes, as well as other properties, such as structure, porosity, and consistency. This result allows inferences regarding the structure and stability of the aggregates. It is known that soils dominated by sand tend to have a weak structure grade, while clay soils have more stable aggregates, indicating a more complex structure [11]. This is explained by the electrochemical charge of the fine particles, which bind with cementing agents such as organic matter, maintaining the bonds and increasing the effective stress [46]. Finally, there is a direct relationship between the flows in the profile. Well-structured soils will have continuous pores, fostering the flow of water and the exchange of gases, while soils with weak structure, such as sands, show water flow in the space between particles [47]. Therefore, soil compressibility, which corresponds to the proportion of a soil mass decreasing its volume when supporting a load, will be lower in coarse soils [46].

Particle size distribution is largely explained by the natural distribution of this property within the soil and is not expected to show great variability. Changes of this property are expected as a result of long-term processes, such as selective erosion or in situ weathering. Nevertheless, the local increase of the clay content affects other properties, such as water retention or organic matter content, that directly affect the mechanical and hydraulic behavior of the soil.

### 4.1.3. Bulk Density (BD)

Due to the tillage practices at this study site, BD is a property that varies within the season [48]. These variations reflect changes in soil structure, which are closely related to total porosity, with the latter being highly sensitive to the content of organic matter (OM) and to the management [46]. This occurs because any load applied to the soil surface is transmitted in three dimensions through the solid, liquid, and gaseous phases. If the air permeability is high enough to allow the immediate deformation of the air-filled pores, soil settlement will primarily affect the flow of water [40].

For a sandy loam textural class, Sandoval, et al. [22] indicated BD values ranging from 1.4 to 1.8 Mg m$^{-3}$, which are higher than the values we observed. On the other hand, it is necessary to consider that sand has a lower compressibility compared to clay soils when subjected to external loads [46], so that depending on their shape, size, and mixture, they can maintain their values relatively stable in front of these mechanical load events [49].

Although the area of high PR presented lower values of BD at the position inside the track, this tendency would not correspond to the expected values due to exposure to strong pressures caused by the passage of machinery in the area under the wheel track. This may be related to the plowing that loosens the soil, inverting and releasing compacted areas. For the case of the high-PR site, which has a higher sand content, the first pass of the machinery generates its final settlement [40], preventing the subsequent rearrangement of particles; meanwhile, in places without machinery crossing (−M), the particles continue to settle with the successive cycles of wetting and drying generated by the irrigations [50]. In the sense of the time after the soil tillage, Osunbitan, et al. [48] reported constant BD increases in different tillage systems after soil preparation, reaching values up to 55% higher after a few weeks.

Finally, to reduce the effect of the pressure exerted by the constant use of machinery for tillage, Horn and Fleige [41] recommend that this must be adjusted to the load capacity of soil and that the wheel load does not exceeds 3.3 Mg, a condition that is generally fulfilled in maize-production systems in Chile.

### 4.1.4. Soil Organic Matter (OM)

Soil organic matter has been applied as a wide term to describe all materials of biological origin in the soil [51]. From the point of view of carbon fluxes, agriculture is defined as the anthropic manipulation of carbon through the absorption, fixation, emission, and transfer of C between the different deposits [52], and its main effect is the emission of $CO_2$ into the atmosphere due to the microbiological combustion of the OM, together

with the stratification of organic waste by tillage systems [53]. However, the long-term incorporation of organic matter to the soil reduces erosion and air pollution by stabilizing the structure and improving the mechanical and hydraulic behavior to disturbances [52].

An OM content lower than 2.5% is considered low, leading to soils with poor biological and physical conditions for Central Chile, according to Ortega and Díaz [54]. Despite this, the values are within the ranges expected for Inceptisols [52], which indicates that repeated management tillage and stubble incorporation has not diminished this property at the time of the study.

The lower OM content in the high-PR areas indicates that there is a loss of OM affecting the fertility and erodibility of the soil, and different results were obtained even with similar soil management.

### 4.2. Hydraulic Functionality of the Soil

4.2.1. Unsaturated Hydraulic Conductivity ($K_h$)

The observed higher values of $K_h$ in the subsoil relative to the subsoil can be explained by the higher amount of sand in depth. There can also be a reduction of $K_h$ in the topsoil due to an initial clogging of the pores as a result of the dispersion of the soil particles due to low stability in water. In addition, plowing contributes to a complete homogenization of the $A_p$ horizon, increasing the total porosity of the porous system but not the continuity [55], and reducing the $K_h$ due to a discontinuity with the subsurface macroporosity [56].

The subsoil had higher $K_h$ values in both PR conditions, thus reflecting a primary effect of compaction by tillage and due to machinery transit. In this regard, Seguel, et al. [9] indicate that, when comparing the effect of ploughing on K, it presents an effect that is prolonged throughout the season, while the effect of machinery passage is not significant in the same season and requires repeated passes to affect K, as occurs in no-tillage systems.

4.2.2. Saturated Hydraulic Conductivity ($K_{sat}$)

The difference in the $K_{sat}$ between high PR and low PR and between topsoil and subsoil can be explained by the same reason as in the range of unsaturated pores, but it is enhanced by the exclusion of the effect of the water tension. The lack of differences in $K_{sat}$ between frequently crossed (+M) or non-crossed (−M) sites in both PR zones may be due to a homogenization of the impact of the labors, which would result from a structural deterioration due to excessive tillage.

The high homogeneity between the two zones and treatments could be due to seasonal plowing management. This is valid, as proposed by Bhattacharyya, et al. [57], who concluded that the conventional tillage systems cause a rupture of the compacted areas, generating an increase of the total porosity of the soil, but a decrease of the amount of macropores, besides its stability and continuity in comparison to the no-tillage systems. This could explain that the results obtained for both high- and low-PR areas generally classify as high-$K_{sat}$ soils. In terms of its permeability, the study site rates as moderately fast, according to Schoeneberger, et al. [39]. These results agree with previous studies that have been carried out at the same site, where it has been determined that the soils show a moderate-to-very-fast $K_{sat}$ [58].

The obtained variability of K in time and space is in the expected range. This property varies considerably under conventional tillage, due to the ploughing of the soil, which coincides with the maximum value of K and decreases during the growing season due to the settlement of the soil particles and formation of structure [59]. In this sense, Strudley, et al. [34] recommend performing temporary/annual measurement campaigns to clarify short-term behaviors, since there are multiple factors that influence and determine the behavior of soil hydraulic conductivity (such as tillage, soil compaction, irrigation, waste management, crop type, climate, soil texture and organic matter content, topography, biotic activity, etc.).

4.2.3. Water Repellency Index (R)

The values of R were considered low [14] and were related to the poor water stability of the aggregates. Together with the silt content (30–40%, Table 1), it could indicate a tendency toward the formation of superficial crusts by a dispersion of aggregates when rapidly wetting [60]. This finding can be explained by the fact that a low repellency is necessary to ensure good stability against wetting events [61]; otherwise, it can cause an increased runoff that can lead to erosion processes and finally a decrease in maize yields.

In the case of excessive repellency or heterogeneous distribution of hydrophobicity, a change in the hydraulic characteristics of the soils would have been expected, leading to the appearance of preferential flows. Fuentes, et al. [58] studied the preferential flow of soluble nitrogen forms in the soil at the same study site, demonstrating that the movement of this element is intrinsically related to that of water and the considerable presence of textural porosity, which is abundant in the soil's coarse texture. In this regard, these authors detected preferential flows, which are not necessarily attributed to phenomena of hydrophobicity, but also implying an environmental risk due to the potential of contamination of underground water.

When exploring the dependence of R with the other properties analyzed, the stepwise regression in both directions indicates clays as the best predictor ($R = 8.809 \times clay + 0.9503$), with statistical significance for this linear model analyzed according to ANOVA. The direct relationship of these properties is in contrary to the findings of Urbanek, et al. [13], indicating that, in coarse textured soils, R tends to be greater. This could eventually be explained by the greater reactivity of clays that retain organic substances. However, OM did not present any kind of correlation or dependence with the other analyzed properties, which could lead to the fact that the cause of this correlation is due to internal characteristics or mineralogy of the clays present in the site, and this should be addressed in future studies.

4.2.4. Correlation between Hydraulic Conductivity (K) and Hydrophobicity (R)

The dependence of $K_{sat}$ with respect to the independent variable R (Figure 7) was evaluated. It is known that C content in soils correlates positively with hydraulic conductivity, because it improves the soil structure [62]. However, the presence of hydrophobic substances in soils can considerably reduce the infiltration processes, and, depending on the magnitude of the repellency, surface runoff processes can be accelerated. Moreover, it was found that even the growth of the plants can be affected by changed surface-water flows [63].

Although the OM contents of the study site are not high (Figure 5), their presence may be influenced by water repellency, since this is a phenomenon that is related to the specific surface occupied by the coating of the particles. Coarse texture classes will be more affected by hydrophobicity than fine texture classes [36].

In soils dominated by sands, as in the present study, with a lower specific surface than in soils dominated by clay, the generation of a hydrophobic surface will have an impact on a greater proportion of particles than for a soil with a fine texture, where the contact surface is up to three times greater [14]. The correlation between both properties was determined by using the mean values of the replicates of each of the treatments of both zones (high and low P). There is an inverse linear dependence between these variables (Figure 7). In addition, it is observed that the zone of low PR presents the highest values of water repellency, a condition that would explain the lower values of K with respect to the site of high PR.

It is possible that the laboratory method used to determine the presence of hydrophobicity has not been sufficiently sensitive, so initially the R index did not explain the problems associated with the inflow of water into the profile. However, even when the R index detected a hydrophilic condition of the study soil, small variations have a direct effect on the $K_{sat}$, indicating that there would be particle dispersion processes due to high wettability. In other words, at the high-PR site, the effect of the higher sand content would be dominate in relation to the low-PR site.

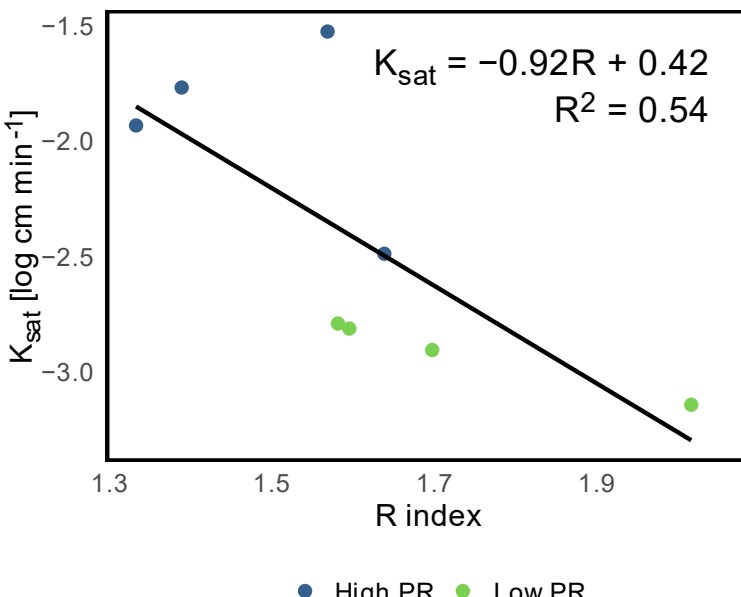

**Figure 7.** Hydraulic conductivity (Log K) as a function of the repellency index (R).

In the context of climate change, with longer periods of drought, it becomes vital to clarify the implications that management has, especially in a resource as closely linked to climate as the soil. In this sense, it becomes relevant to evaluate the critical value of R (> 1.95) for specific local conditions and its temporal distribution along the season to reduce environmental degradation processes and improve conditions for maize cultivation, making optimal use of soil and water.

## 5. Conclusions

For an Inceptisol in Central Chile under traditional intensive tillage, the effect of machinery traffic evaluated as penetration resistance (PR) in places with frequent crossing of machinery (+M) and places without crossing (−M) was not observed. It shows a homogeneous distribution, possibly mitigated by annual tillage. However, a hardening at depth was observed, reflected as an increase in bulk density and penetration resistance of the soil. It is particularly noteworthy that the distribution of PR is associated with the clay content of the soil, possibly associated with the mayor water and nutrient retention capacity that these particles have, which favors a mayor accumulation of organic matter.

When analyzing soil functionality through hydraulic properties, evaluated as hydraulic conductivity (K) and soil water repellency (R) and associated with the movement of water in the soil, we see homogeneous values for the different treatments. However, the results indicate a reduction in K in the low-PR zone, which can be explained by the higher clay content and OM. When exploring the distribution in depth, there was an increase in K the subsoil, which can be attributed to the conservation of coarse porosity resulting from structuring and biological activity, while on the surface, it is destroyed by tillage. The effect of the passage of machinery (−M/+M) was not affecting the K.

Findings on water repellency (R) indicated that there were no significant differences in hydrophobicity at the study site. This result contrasts with the observed correlated trend between hydraulic conductivity and the R index. This may be due to the sensitivity of the method used, which was not high enough to determine a sufficient degree of soil hydrophobicity. This could be explained by the presence of surface runoff and the problems of water infiltration in the profile identified by farmers. It might be interesting to study this phenomenon more thoroughly and to conduct measurements better distributed throughout the season. Moreover, R shows a dependency on the clay content that is not associated with OM or other texture fractions, and this could be explained by internal characteristics or mineralogy of the clays present in the site. This result should be addressed in future studies.

**Author Contributions:** Conceptualization, O.S. (Oscar Seguel), O.S. (Osvaldo Salazar) and N.R.-M.; methodology, O.S. (Oscar Seguel) and N.R.-M.; software, N.R.-M.; formal analysis, N.R.-M.; investigation, C.S. and N.R.-M.; resources, O.S. (Oscar Seguel) and O.S. (Osvaldo Salazar); data curation, C.S.; writing—original draft preparation, N.R.-M.; writing—review and editing, S.S. and T.S.; visualization, N.R.-M.; supervision, O.S. (Oscar Seguel) and O.S. (Osvaldo Salazar); project administration, O.S. (Osvaldo Salazar); funding acquisition, O.S. (Osvaldo Salazar). All authors have read and agreed to the published version of the manuscript.

**Funding:** This research was partially funded by FONDECYT de Iniciación 2011 grant no. 11110464.

**Institutional Review Board Statement:** Not applicable.

**Informed Consent Statement:** Not applicable.

**Data Availability Statement:** The data presented in this study are available on request from the corresponding author.

**Acknowledgments:** The authors thank the Departamento de Ingeniería y Suelos of the Universidad de Chile and the Cooperativa Intercomunal Campesina de Peumo (COOPEUMO) for supporting this study. To Eduardo Figueroa for his permission to work at the study site.

**Conflicts of Interest:** The funders had no role in the design of the study; in the collection, analyses, or interpretation of data; in the writing of the manuscript; or in the decision to publish the results.

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
