# Peer review of "Variability of Hydraulic Properties and Hydrophobicity in a Coarse-Textured Inceptisol Cultivated with Maize in Central Chile"

_soilsystems, doi:10.3390/soilsystems6040083_

Round 1
Reviewer 1 Report
I appreciate the interesting location and the pedological assessment. The paper has several significant formal flaws. Please use only SI units in the article (do not use "Mg"; or "cm"). The text is not properly divided into chapters. Evaluation of the influence of mechanization is unacceptable. Their technical parameters determine the influence of tires on soil properties. The design of your observation does not allow this! The terms "in-the-wheel-track (IT)" and "outside-the-wheel-track" (OT) are unacceptable. You can replace "places with frequent crossing" and "places without crossing".
Abstract:
· It is not clear what new things you have discovered.
· Line 12: "intensive tillage" - replace "intensive"
Introduction:
· Line 34: "other Mediterranean regions", what do you mean by "other"? "Mediterranean" - this term is used to refer to the region around the Mediterranean Sea. Consider replacing this term in both the text and the title. It will be good to avoid confusion.
· Line 75 "excessive tillage" - how do you understand the term "excessive"?
· Define hypotheses. It is not clear what you expect. The title shows that you want to evaluate the variability of selected properties. Please include it in the hypotheses. Considering your methodology, goal (i) is unacceptable.
Materials and Methods:
· Any information about technology and tires is missing.
· Lines 109 – 110: It is necessary to describe the entire technology of growing corn.
· Line 112: You cannot prove this. If you don't use GPS navigation and even then, it's questionable.
Results and Discussion:
· I recommend dividing it into two chapters
· The text blurs the differences between your results and the quoted data.
Conclusions:
· Write only your most important findings!
· Line 511 – 512: "This directly affects soil functionality, limiting the ability of the roots to reach water and nutrients, reducing maize yields, but this effect could be eliminated through deep ploughing." - are these your results?
English errors:
Line 40: ..making difficult an straightforward comparisons.. A straightforward
Reviewer 2 Report
The reviewed manuscript concerns the variability of the basic soil properties (mechanical composition, soil bulk density, organic mater content) as well as soil hydraulic conductivity and water repellency index. Measured values of soil penetration resistance were used for establishing four treatments (measuring points) where soil properties were measured. The study was conducted on coarse-textured Inceptisol on a farm in central Chile, in a fallow-maize rotation under conventional tillage. The results of measured data were analyzed using statistical methods (kriging, ANOVA, regression). The reviewed manuscript is thematically appropriate for “Soil Systems” journal. The manuscript is well written and the overall layout of the work is correct and legible. The introduction provides sufficient background and supports the research topic of the study. The description of the research methodology used is accurate, however requires some additional information. Discussion is well written and conclusions presented are result from the conducted research and analysis
Comments (suggestions):
In Fig.1 and 2 please add scale of the map. On Fig. 2 please mark location “four treatments”
Please clarified how many samples were used for determination of mechanical composition, BD, OM and R. Line 128 – 32 samples for hydraulic conductivity, how many samples were used for other characteristics?
Line 185 – please add reference for Surfer 10, please specified what type of kriging was used for interpolation.
Line 207 – please add reference for R package
Please specified the median value of PR which was used for division of PR into high and low class. I cannot find what does it mean high and low PR values.
For me caption of Fig. 2 is not clear. I do not see “Average areas”
What type ANOVA was used for the analysis of the data? One-way? Why not 3-way (zone, treatment, position)?
In table 1 results of ANOVA analysis are presented as average values and SD. Why not 95% interval? The detection of the existence of the significant differences in Table 1 is not very clear to me (some letters are italic bold, some not)
In Fig 6 – symbols OWT and WT should be explained (or changed)
In my opinion “Soil water pressure” should be presented as negative values (suction)
When you citated paper whit 3 and mor authors please use Author et al. (for example lines 312, 331 and more, please check this).
In Keywords instead of hydrophobicity please use repellency index
Round 2
Reviewer 1 Report
I thank the authors for their cooperation.